# Herpes Simplex Virus 1 Infection of Human Brain Organoids and Pancreatic Stem Cell-Islets Drives Organoid-Specific Transcripts Associated with Alzheimer’s Disease and Autoimmune Diseases

**DOI:** 10.3390/cells13231978

**Published:** 2024-11-29

**Authors:** Jonathan Sundstrom, Emma Vanderleeden, Nathaniel J. Barton, Sambra D. Redick, Pepper Dawes, Liam F. Murray, Meagan N. Olson, Khanh Tran, Samantha M. Chigas, Adrian R. Orszulak, George M. Church, Benjamin Readhead, Hyung Suk Oh, David M. Harlan, David M. Knipe, Jennifer P. Wang, Yingleong Chan, Elaine T. Lim

**Affiliations:** 1Department of Medicine, Division of Innate Immunity, University of Massachusetts Chan Medical School, Worcester, MA 01605, USA; 2Department of Molecular, Cell and Cancer Biology, University of Massachusetts Chan Medical School, Worcester, MA 01605, USA; 3Department of Neurology, University of Massachusetts Chan Medical School, Worcester, MA 01605, USA; 4NeuroNexus Institute, University of Massachusetts Chan Medical School, Worcester, MA 01605, USA; 5Department of Genomics and Computational Biology, University of Massachusetts Chan Medical School, Worcester, MA 01605, USA; 6Department of Medicine, Diabetes Center of Excellence, University of Massachusetts Chan Medical School, Worcester, MA 01605, USA; 7Program in Molecular Medicine, Diabetes Center of Excellence, University of Massachusetts Chan Medical School, Worcester, MA 01605, USA; 8Graduate Program in Biochemistry & Molecular Biotechnology, University of Massachusetts Chan Medical School, Worcester, MA 01605, USA; 9Graduate Program in Neuroscience, University of Massachusetts Chan Medical School, Worcester, MA 01605, USA; 10Graduate Program in Immunology and Microbiology, University of Massachusetts Chan Medical School, Worcester, MA 01605, USA; 11Department of Genetics, Blavatnik Institute, Harvard Medical School, Boston, MA 02115, USA; 12Wyss Institute for Biologically Inspired Engineering, Harvard University, Boston, MA 02115, USA; 13ASU-Banner Neurodegenerative Disease Research Center, Arizona State University, Tempe, AZ 85281, USA; 14Department of Microbiology, Blavatnik Institute, Harvard Medical School, Boston, MA 02115, USA

**Keywords:** stem cell islets, cerebral organoids, type 1 diabetes, Alzheimer’s disease, innate immune, neurodegenerative diseases, autoimmune diseases, acyclovir, herpes simplex virus 1

## Abstract

Viral infections leading to inflammation have been implicated in several common diseases, such as Alzheimer’s disease (AD) and type 1 diabetes (T1D). Of note, herpes simplex virus 1 (HSV-1) has been reported to be associated with AD. We sought to identify the transcriptomic changes due to HSV-1 infection and anti-viral drug (acyclovir, ACV) treatment of HSV-1 infection in dissociated cells from human cerebral organoids (dcOrgs) versus stem cell-derived pancreatic islets (sc-islets) to gain potential biological insights into the relevance of HSV-1-induced inflammation in AD and T1D. We observed that differentially expressed genes (DEGs) in HSV-1-infected sc-islets were enriched for genes associated with several autoimmune diseases, most significantly, T1D, but also rheumatoid arthritis, psoriasis, Crohn’s disease, and multiple sclerosis, whereas DEGs in HSV-1-infected dcOrgs were exclusively enriched for genes associated with AD. The ACV treatment of sc-islets was not as effective in rescuing transcript perturbations of autoimmune disease-associated genes. Finally, we identified gene ontology categories that were enriched for DEGs that were in common across, or unique to, viral treatment of dcOrgs and sc-islets, such as categories involved in the transferase complex, mitochondrial, and autophagy function. In addition, we compared transcriptomic signatures from HSV-1-infected sc-islets with sc-islets that were infected with the coxsackie B virus (CVB) that had been associated with T1D pathogenesis. Collectively, this study provides tissue-specific insights into the molecular effects of inflammation in AD and T1D.

## 1. Introduction

Inflammation has been shown to be one of the key factors for disease pathogenesis or progression in several common, complex diseases, such as Alzheimer’s disease (AD) and type 1 diabetes (T1D). Viral-induced inflammation due to the double-stranded DNA herpes simplex virus 1 (HSV-1) has been associated with AD risk [1,2,3,4], whereas viral-induced inflammation due to the single-stranded RNA coxsackie B virus (CVB) has been associated with T1D risk [5,6]. In parallel, embryonic stem cell or induced stem cell-derived models, such as 3D cerebral organoids and stem cell-derived islets (sc-islets), have been shown to be promising in vitro human cell systems for studying the molecular and cellular effects of viral-induced inflammation that are relevant for common, complex diseases, especially when primary tissue from living human donors may not be easily accessible [4,7,8,9].

Previously, we found that HSV-1 infection in dcOrgs led to several molecular and cellular phenotypes associated with AD neuropathology, including enrichment of differentially expressed genes (DEGs) in AD-associated genes [4]. On the other hand, viral infections in dcOrgs, by using the single-stranded RNA virus influenza A (IAV), did not result in these AD-associated phenotypes. This suggests that AD pathology can be activated by herpesviruses, such as HSV-1. Interestingly, we also observed that HSV-1-infected dcOrg samples that were treated with the anti-viral drug acyclovir (ACV) had differential expression in transcripts associated with autoimmune diseases, such as T1D and rheumatoid arthritis (RA). This indicates that the presence of HSV-1 viral constructs may be sufficient to elucidate host inflammatory programs that are shared among different autoimmune diseases.

As such, we sought to test the following hypothesis: Can HSV-1 infection in sc-islets similarly lead to an enrichment of DEGs in AD-associated genes? If the hypothesis is true, the results will indicate that shared host inflammatory programs due to HSV-1 infection are driving disease-associated mechanisms in AD, regardless of the in vitro human system (sc-islets or dcOrgs). On the other hand, if the hypothesis is not true, the results will indicate that there might be cell type-specific or organoid-specific host inflammatory programs that are driving disease-associated mechanisms in AD and T1D.

We further explore the global transcriptomic profile of HSV-1-induced inflammation in sc-islets and compare RNA sequencing (RNA-seq) data from HSV-1 infection (with and without ACV treatment) of sc-islets and dcOrgs to understand the perturbed gene networks and pathways that are in common or specific to each stem cell-derived system. Through our cross-organoid and cross-disease study, we aim to understand the specificity, efficacy, and potential off-target effects of acute inflammation and anti-viral treatment, which can physiologically affect multiple organs in humans.

## 2. Materials and Methods

### 2.1. Cerebral Organoid Differentiation and Dissociation

We used a single donor, induced pluripotent stem cell (iPSC) line for differentiating dcOrgs. The donor line was obtained from the Harvard Personal Genome Project (HUID: hu43860C, PGP#: 1), which is a unique resource for sharing genomes, phenotypes, and cell lines for research [4,7,10,11,12,13,14,15,16]. 3D cerebral organoids were differentiated using a previously published protocol by Lancaster et al. [4,7,17]. Embryoid bodies comprising ~9000 cells each, were formed in 96-well ultra-low attachment plates (Corning Life Sciences, Tewksbury, MA, USA Cat# 3474) and transferred to 24-well ultra-low attachment plates (Corning Life Sciences, Tewksbury, MA, USA Cat# 3473) with 500 μL of neural induction media on Day 6 as described [4,7,17]. Subsequently, the organoids were embedded in 40 μL of Matrigel (Corning Life Sciences, Tewksbury, MA, USA Cat# 354234) four days later and placed in 2 mL of differentiation media on an orbital shaker at 90 rpm for differentiation. After 2–4 months of differentiation, we dissociated the 3D cerebral organoids using our previously reported protocol [4,16]. Briefly, the organoids were washed in cold 1 × DPBS (Life Technologies, Carlsbad, CA, USA Cat# 14190250) for 10 min and then incubated in 500 μL of 0.25% Trypsin-EDTA (Life Technologies, Carlsbad, CA, USA Cat# 25300054) for 15 min to dissociate the organoids. Dissociated cells were passed through a 30 μm cell strainer (Miltenyi Biotec, Auburn, CA, USA Cat# 130-098-458) prior to replating on Matrigel-coated plates. Following dissociation, cells were passaged for at least a month to allow recovery. The cerebral organoids used in our study had been reported to harbor microglia (that differentiated from the mesoderm lineage) innately within these organoids, thus providing a good human in vitro model for studying innate immune responses in the context of other cell types found in the central nervous system.

### 2.2. Sc-Islet Differentiation

The luciferase expressing the ES cell line HVRDe008-A-1 (GAPluc) [18] was maintained and differentiated using protocol v8 as described [19]. The glucose-stimulated insulin secretion index calculated by dividing the amount of insulin secreted following incubation in high glucose (20 mM) by that following incubation with basal glucose (2.8 mM), was 1.5 in Stage 6, on Day 15. HUES8-derived sc-islets can be reproducibly generated from a single genetic background (HUES8 is among the first lines approved by NIH in response to a 2009 executive order enabling more hESC lines to be added to the list of those approved for federally-funded research). They are accessible to other investigators. The sc-islets are derived from endoderm and, thus, lack macrophages that could confound innate immune responses in primary human islets.

### 2.3. HSV-1 Viral Infection and ACV Treatment of dcOrgs

In our work, we used the HSV-1 K26GFP KOS strain that was generously provided by David Knipe and Prashant Desai [20,21,22]. The virus has a green fluorescent protein (GFP) fused in frame with the viral UL35 ORF, such that it will express a VP26-GFP fusion protein during late infection. Dissociated cells dcOrgs were infected according to our previously reported protocol [4]. Briefly, 1 × 10^6^ dcOrgs per well were plated onto Matrigel-coated 6-well plates, and we performed HSV-1 inoculation for 1 h using a multiplicity of infection (MOI) of 2. For the ACV-treated dcOrgs, 200 μM of ACV was added to HSV-1-infected dcOrgs immediately. After an hour, inoculation media were removed, the dcOrgs were rinsed with 1 × DPBS, and differentiation media were added to the dcOrgs for 23 h. For the ACV-treated dcOrgs, 200 μM of ACV was added to the differentiation media for the next 23 h. We confirmed that the dcOrgs were infected using a fluorescence microscope the next day.

### 2.4. Viral Infection and Anti-Viral Drug Treatment of Sc-Islets

We used the same HSV-1 K26GFP KOS strain for viral infections with the sc-islets. 500,000 cells per well were plated in low attachment 24-well plates and inoculated with HSV-1 for 1 h using a MOI of 5 in incomplete CMRL 1066 media. We chose to use a MOI of 5 in our sc-islet model system to ensure that >50% of the cells would be infected and that we would observe significant transcriptomic perturbations after 48 h. We would like to note that viral inflammation-induced disease pathology in humans is likely to be driven by low-dosage chronic reactivation of latent viruses over several years. 200 μM of ACV was added to some of the HSV-1-infected cells immediately. After an hour, inoculation media were removed and replaced with CMRL media supplemented with 10% fetal bovine serum, (Life Technologies, Carlsbad, CA, USA Cat# A5670801), 1% penicillin-streptomycin solu-tion (Life Technologies, Carlsbad, CA, USA Cat# 10378016), and 1% GlutaMAX (Life Technologies, Carlsbad, CA, USA Cat# 35050061). 200 μM of ACV was added to the cells that were infected with HSV-1 and treated with ACV. The cells were harvested after 48 h.

We had also used influenza A (IAV) strain A/Puerto Rico/8/1934(H1N1) that was purchased from Charles River Laboratories (Wilmington, MA, USA) [23], with a final HA titer per 0.05 mL of 131,072 and the EID50 titer per mL was 10^9.8^. The CVB strain used in our study was strain JVB (ATCC Cat #VR-184) [8].

### 2.5. RNA Extraction and Library Preparation

We fixed the cells with 4% paraformaldehyde and extracted RNA using the PureLink^TM^ FFPE RNA Isolation kit (Life Technologies, Carlsbad, CA, USA), followed by DNase I treatment. We shipped all total RNA samples overnight on dry ice to Psomagen (Rockville, MD, USA) for quantification, ribosomal RNA depletion, and library preparation, followed by 151 bp paired-end sequencing, with a total of 40 million reads on a NovaSeq6000 (Illumina, San Diego, CA, USA).

### 2.6. RNA-Seq Alignment and Analyses

As previously described [4], we performed pre-processing using FastQC [24], Trimmomatic v0.39 [25], HISAT2 v2.2.1 [26] and SAMtools v1.9 [27]. For reference alignment, we used a concatenated GRCh37 human reference genome and human alphaherpesvirus 1 Kos strain reference genome (GenBank: JQ673480.1). We performed expression quantification using StringTie2 v1.3.6 [28]. Pearson’s correlations r were calculated using the fragments per kilobase of exon per million mapped fragments (FPKM) values between sample pairs. Schematic figures were created using BioRender. All other figures were created using R v4.3.2 (https://www.r-project.org/, accessed on 13 November 2023) or Python v3.12.0 (https://www.python.org/, accessed on 13 November 2023).

Differentially expressed genes (DEGs) were defined as having adjusted *p* ≤ 0.05. For the rescued analyses, we performed conditional analyses by identifying the DEGs between HSV-1-infected versus HSV-1-infected and ACV-treated samples. Next, DEGs between HSV-1-infected versus uninfected samples (adjusted *p* ≤ 0.05), which had the same fold change direction (up-regulated or down-regulated), were classified as rescued.

Pathway analyses on DEG lists were performed using g:Profiler with g:SCS multiple testing correction and a significance threshold of 0.05 [29]. DEGs were queried against the Gene Ontology knowledgebase [30]. We excluded pathways with electronic GO annotations and only included annotated genes in the domain scope. Pathways surpassing the significance threshold were filtered to only include pathways with a term size of less than 500 and more than 15 genes.

### 2.7. Statistical Tests

Statistical tests used in our study were conducted in R for Pearson’s correlations, and Fisher’s exact tests. For the gene set enrichment analyses, we downloaded the gene lists from the NHGRI-EBI GWAS catalog in April 2022 [31] and conducted the analyses using GSEA v4.3.2 [32,33], by using the DEGs ranked by absolute *log_2_* fold change in decreasing order. We used 10,000 permutations and a weighted enrichment statistic as additional parameters.

## 3. Results

### 3.1. High Correlations Were Observed Across RNA-Seq Data from Sc-Islets

Previously, we detected high correlations in RNA-seq data from different dcOrg replicates using our approach [4], and sought to similarly evaluate the correlations in RNA-seq data from different sc-islet replicates. We observed that using an MOI of 5, ~40% of islets were infected after 24 h of infection, and ~80% of sc-islets, including insulin-positive cells, were infected after 48 h of infection (Figure 1A). Subsequently, we harvested sc-islets after 48 h of infection for RNA extraction and sequencing. From the RNA-seq data on the sc-islets, we found that 70 ± 11% of all transcripts in the triplicate HSV-1-infected sc-islets were viral transcripts, and 1.5 ± 0.4% of all transcripts in the triplicate ACV-treated (HSV-1-infected and ACV-treated) sc-islets were viral transcripts. Previously, we observed that using a MOI of 2 resulted in >50% of infected dcOrgs after 24 h of infection, and 60–81% of all transcripts in HSV-1-infected dcOrgs were viral transcripts, whereas 0.15–3% of all transcripts in treated dcOrgs were viral transcripts, demonstrating that ACV treatment worked to reduce viral transcript abundance in treated dcOrgs [4].

Principal components analyses (PCA) showed that 87.8% of the variation was captured by the first principal component (PC1) and that the infection status of the sc-islets correlated with PC1 (Figure 1B). Pearson’s correlations (r) were high between replicate samples with the same condition (r = 0.85–0.99), whereas correlations were lower between uninfected versus infected samples (r = 0.36–0.59), treated versus infected samples (r = 0.47–0.87) and treated versus uninfected samples (r = 0.72–0.96), as shown in Appendix A, indicating that there were detectable transcriptomic perturbations due to infection or ACV treatment in sc-islets.

### 3.2. Differentially Expressed Genes in HSV-1-Infected Versus Uninfected Sc-Islets Were Enriched for Autoimmune Disease-Associated Genes

We computed the list of differentially expressed genes (DEGs) from HSV-1-infected and uninfected sc-islets (Appendix A). Gene ontology analyses revealed that DEGs from HSV-1-infected versus uninfected sc-islets were enriched in processes, such as p38 MAPK cascade, whereas DEGs from HSV-1-infected versus HSV-1-infected and ACV-treated sc-islets were enriched in processes related to sarcomere, myofibril, and contractile fiber (Appendix A). We tested if DEGs were enriched for any lists of genes associated with 21 common diseases and traits from the GWAS Catalog (Figure 1C, Appendix A). We found enrichment of the DEGs for several autoimmune disease-associated genes, including for T1D, *p* = 5.6 × 10*^−^*^3^; RA, *p* = 0.02; psoriasis (PSO), *p* = 0.026; Crohn’s disease (CD), *p* = 0.045; as well as for the neurodegenerative and autoimmune disease, multiple sclerosis (MS), *p* = 0.022. Unlike our prior results from dcOrgs, we did not observe enrichment of the DEGs from HSV-1-infected versus uninfected sc-islets for AD-associated genes (*p* = 0.69), indicating that HSV-1 infection leading to transcriptomic perturbations associated with AD was specific to dcOrgs.

There were six DEGs in common across genes associated with T1D, RA, PSO, CD, and MS: IRF1, CD226, RUNX3, KEAP1, ZMIZ1, and ATXN2. These six genes are involved in host response to viral infection, such as cytokine production and oxidative stress response, leading to cell death. However, the GSEA results did not change significantly after removing these six DEGs (T1D, *p* = 7 × 10*^−^*^3^; RA, *p* = 0.039; PSO, *p* = 0.025; CD, *p* = 0.062; and MS, *p* = 0.034), indicating that the enrichment of DEGs was likely to be driven by several genes that were exclusively associated with each autoimmune disease. The T1D-specific genes include the insulin gene (INS), which was significantly decreased in HSV-1 infected sc-islets (log_2_ fold change = −2.7, adjusted *p* = 4.1 × 10*^−^*^3^). Additional T1D-specific genes are shown in Appendix A.

### 3.3. ACV Treatment of HSV-1-Infected Sc-Islets Did Not Rescue Human Transcript Differential Expression

If ACV treatment was effective in rescuing human transcript expression associated with these autoimmune diseases in HSV-1-infected sc-islets, we would expect DEGs from HSV-1-infected versus HSV-1-infected and ACV-treated sc-islets (Islet_Inf versus ACV) to be similarly enriched in these autoimmune disease-associated genes. However, when we conducted gene set enrichment analysis (GSEA) using the Islet_Inf-vs-ACV DEGs, we observed no enrichment for any of these six autoimmune disease-associated genes (T1D, *p* = 0.17; RA, *p* = 0.75; PSO *p* = 0.45; CD, *p* = 0.37; MS, *p* = 0.67).

Alternatively, if ACV treatment was not effective in rescuing human transcript expression associated with these autoimmune diseases, we would expect to observe an enrichment of DEGs from ACV-treated versus uninfected sc-islets (Islet_ACV versus Ctrl). For this analysis, GSEA showed no enrichment of the DEGs for these select autoimmune disease-associated genes (T1D, *p* = 0.095; RA, *p* = 0.74; PSO, *p* = 0.24; CD, *p* = 0.25; MS, *p* = 0.052). These results suggest that the 200 μM ACV treatment to inhibit HSV-1 DNA replication in sc-islets may not be sufficient to rescue human transcript expression in genes associated with T1D and potentially other autoimmune diseases.

These results are in contrast to our prior results with ACV treatment on HSV-1-infected dcOrgs, where we observed a dosage-dependent effect of ACV treatment on rescuing perturbations in AD-associated transcripts [4]. We previously found that a set of ACV-treated HSV-infected dcOrgs with a low abundance of viral transcripts (0.15–0.17% of all transcripts mapped to viral transcripts) showed an enrichment of DEGs associated with AD when compared to HSV-1-infected dcOrgs, which indicated a rescue of human transcript expression associated with AD. A second set of ACV-treated, HSV-infected dcOrgs with a higher abundance of viral transcripts detected (2.4–3% of all transcripts mapped to viral transcripts) did not show an enrichment of DEGs associated with AD when compared to HSV-1-infected dcOrgs, which indicated a lack of rescue. However, we observed an enrichment of DEGs associated with AD when we compared the second set of ACV-treated dcOrgs with uninfected dcOrgs, which indicated that there was a dosage-dependent effect of ACV treatment on rescuing transcript perturbations in AD-associated genes. These results also suggest that the 200 μM ACV treatment to inhibit HSV-1 DNA replication in sc-islets may not be sufficient to rescue human transcript expression in genes associated with T1D and potentially other autoimmune diseases.

Among the top 50 DEGs from sc-islets, ACV treatment had rescued the expression of some genes (e.g., BACE1) to levels that were more similar to uninfected sc-islets (Figure 1D). However, ACV treatment did not rescue the expression of most genes to levels that were more similar to uninfected sc-islets. These results are consistent with our observation that ACV treatment did not rescue the transcript expression in most genes.

### 3.4. GSEA Analyses of Previously Published HSV-1-Infected Cerebral Organoids Across GWAS-Associated Genes Indicate Differences in 3D Versus 2D Cultures

Previously, there were two studies that generated RNA-seq data from HSV-1-infected 3D cerebral organoids [34,35]. It was reported that HSV-1 infection in 3D cerebral organoids led to a more attenuated type 1 interferon (IFN-I) response than 2D cultures of non-neural or neural cells, including dissociated cells from 3D cerebral organoids (that we termed as dcOrgs) [35]. We ran GSEA analyses using both datasets and observed modest enrichment of the DEGs with AD-associated GWAS genes (Figure 1E). These results are unlike our observations using the DEGs from 2D dcOrgs [4], and the differences might be due to attenuated IFN-1 induction in 3D cerebral organoids compared to 2D dcOrgs [35].

### 3.5. Analyses of DEGs That Are in Common Between HSV-1-Infected dcOrgs and Sc-Islets Revealed Genes Involved in Amyloid-Beta Clearance, RNA Metabolic, Processing and Stability, as Well as Mitochondria Function

We further compared the RNA-seq data from the islets to our previously generated dcOrgs RNA-seq data [4]. In our dcOrgs data, we had two sets of replication experiments for each comparison (dcOrgs1 and dcOrgs2), and we compared the DEGs in sc-islets with dcOrgs1 or sc-islets with dcOrgs2 that were significant (adjusted *p*-value *≤* 0.05) and had log_2_ fold changes in the same direction (up-regulated or down-regulated).

We found that 45.2% and 45.8% of the genes that were DEGs in HSV-1-infected versus uninfected dcOrgs were also DEGs in HSV-1-infected versus uninfected sc-islets, as shown in Figure 2A. More up-regulated DEGs were common to both dcOrgs and sc-islets compared to down-regulated DEGs (50.9% versus 38.8% for dcOrgs1, and 51.5% versus 39.4% for dcOrgs2; odds ratio (OR) = 1.6; Fisher’s exact test *p* < 1 × 10*^−^*^10^ for both replicates; Appendix A).

### 3.6. Proportionally Fewer Common Human DEGs Are Rescued with ACV Treatment of HSV-1-Infected Sc-Islets Compared to dcOrgs

Next, we wondered about the effects of ACV treatment on HSV-1-infected sc-islets versus dcOrgs. We used the DEGs identified from HSV-1-infected and ACV-treated sc-islets or dcOrgs, versus HSV-1-infected sc-islets or dcOrgs for these comparisons. We observed that fewer DEGs were common to both the sc-islets and dcOrgs (21.4% for dcOrgs1 and 22.2% for dcOrgs2, Figure 2B), compared to DEGs in common between HSV-1-infected sc-islets and dcOrgs (Figure 2A). These results indicated that ACV treatment of HSV-1-infected dcOrgs had more organoid-specific effects than HSV-1 infection alone, and therefore, fewer DEGs were attributed to ACV treatment that were in common between dcOrgs and sc-islets.

However, there were still proportionally more up-regulated DEGs than down-regulated DEGs that were in common between treated sc-islets and dcOrgs (26.8% versus 15.9% for dcOrgs1, and 28.8% versus 15.4% for dcOrgs2; OR = 1.9 and 2.2; Fisher’s exact test, *p* < 1 × 10*^−^*^10^ for both replicates, Appendix A).

### 3.7. ACV Treatment of HSV-1-Infected Sc-Islets Did Not Completely Inhibit True-Late Viral Gene Expression

Previously, we found that ACV treatment on our second set of dcOrgs (dcOrgs2) was not as effective as ACV treatment on our first set of dcOrgs (dcOrgs1) [4]. Since ACV primarily inhibits transcript expression of HSV-1 true-late viral genes, we observed significant differences in the ranked expression of true-late viral genes between both sets of dcOrgs but did not observe significant differences in the ranked expression of leaky-late viral genes between both sets of dcOrgs [4]. We compared the ranked expression of leak-late or true-late viral genes between the sc-islets and both sets of dcOrgs (Appendix A) and found that there was a strong correlation in the expression of leaky-late viral genes between sc-islets and dcOrgs1 or dcOrgs2 (*p* = 3.4 × 10*^−^*^4^ and 4.1 × 10*^−^*^4^, respectively). However, the ranked expression of true-late viral genes in the sc-islets was more similar to those in dcOrgs2 (*p* = 7.7 × 10*^−^*^3^) than those in dcOrgs1 (*p* = 0.11). These results indicate that ACV treatment at the dosage and duration used in our study was not as effective in inhibiting true-late viral gene expression in the sc-islets.

### 3.8. Gene Ontology Analyses of DEGs in HSV-1-Infected Sc-Islets and dcOrgs Identified Common Functional Categories Involved in the Transferase Complex, Mitochondrial, and Autophagy Function

Next, we performed gene ontology (GO) enrichment analyses to identify the distinct GO categories that were enriched among the various groups of DEGs. We conducted two different sets of analyses with the ACV-treated cells, as illustrated in Figure 3. To evaluate the effects of ACV treatment, we compared the HSV-1-infected and ACV-treated cells to the HSV-1-infected cells (called “ACV analyses”). To evaluate the conditional effects of ACV treatment, given the effects of HSV-1 infection, we compared the HSV-1-infected and ACV-treated cells versus HSV-1-infected cells for only human transcripts that were differentially expressed in HSV-1-infected cells versus uninfected cells (called “rescued analyses”).

For the ACV analyses, e.g., dcOrgs_only_ACV_up and Islets_only_ACV_up, the comparisons aimed to evaluate the effect of ACV treatment on human transcripts. For instance, if the expression of a gene was up-regulated due to ACV, the HSV-1-infected and ACV-treated samples were compared with the HSV-1-infected samples. For the rescued analyses, e.g., dcOrgs_only_rescued_down and Islets_only_rescued_down, the comparisons aimed to evaluate the conditional effect of ACV treatment, given the effect of HSV-1 infection on human transcripts. For instance, if the expression of a gene was up-regulated due to HSV-1 infection, and the expression of the same gene was down-regulated by ACV treatment, then we termed the gene as a rescued gene.

The GO categories that were up-regulated, both in common with HSV-1-infected sc-islets and HSV-1-infected dcOrgs, were in the transferase complex involved in transferring phosphorus-containing groups (Common_inf_up; Figure 4 and Appendix A; Appendix A). Phosphatases were necessary for viral entry into cells or cell-to-cell spread, and it was reported that phosphorylation can modify regulatory protein activity in infected cells [36,37]. The GO categories that were down-regulated, both in common with sc-islets and dcOrgs, included mitochondrial and autophagy function (Common_inf_down), which are key processes that are critical for innate immune activation but are also reported to be hijacked by viruses to suppress innate immunity [38,39,40].

The GO categories that were up-regulated only in sc-islets included genes that were involved in antigen processing and the presentation of peptide antigen, as well as responses to bacteria (Islets_only_inf_up); this is consistent with prior observations in islets from T1D patients [41,42]. The GO categories that were down-regulated only in sc-islets included axon and dendrite development (Islets_only_inf_down). This is consistent with prior reports, which indicated that impaired insulin signaling can affect dendrite and synaptic regeneration [43,44]. The GO categories that were down-regulated only in dcOrgs included DNA replication, recombination, helicase, and conformation change (dcOrgs_only_inf_down).

We explored the direct effects of ACV treatment on sc-islets and dcOrgs, by comparing the GO categories that were enriched in HSV-1 infection and ACV treatment versus HSV-1 infection on sc-islets and dcOrgs (Figure 3). The DEGs in these categories comprise human genes where the expression was perturbed by ACV treatment (up-regulated or down-regulated). We observed that the categories that were up-regulated only in dcOrgs were involved in cytoskeleton and organelle organization (dcOrgs_only_ACV_up), whereas the categories that were only up-regulated in sc-islets were in receptor complexes (Islets_only_ACV_up), including cytokine receptors (IL23R, IL31RA, and IL5RA) and toll-like receptors (TLR1 and TLR7).

Finally, we explored the conditional effects of ACV treatment on sc-islets and dcOrgs (rescued analyses). We compared the GO categories that were: (1) enriched in HSV-1 infection and ACV treatment versus HSV-1-infection; (2) enriched in HSV-1 infection versus uninfected in sc-islets and dcOrgs. The GO categories that were down-regulated due to HSV-1 infection and rescued by ACV treatment only in sc-islets included gated channel activity and monoatomic ion channel complex (Islets_only_rescued_down). On the other hand, the GO categories that were down-regulated due to HSV-1 infection and rescued only in dcOrgs included mitochondrial and ribosome subunits (dcOrgs_only_rescued_down).

### 3.9. Comparisons of Transcriptomic Perturbations by RNA and DNA Viruses in Sc-Islets and dcOrgs Revealed Similarities in Gene Expression

HSV-1 is a double-stranded DNA virus, whereas CVB is a single-stranded RNA virus that was previously associated with T1D pathogenesis. Previously, Nyalwidhe et al. infected primary or sc-islets with CVB and performed RNA-seq [8,9]. Similarly, we infected dcOrgs with the RNA virus influenza A virus (IAV) and performed RNA-seq [4]. We asked if the global human transcriptomic profiles of RNA viral infections in sc-islets and dcOrgs were more similar to the transcriptomic profiles of an RNA viral infection (CVB) in sc-islets versus a DNA viral infection (HSV-1) in dcOrgs. We found that human transcripts that were up-regulated by CVB infection in sc-islets were more enriched for up-regulated transcripts by IAV infection in dcOrgs versus HSV-1 infection in dcOrgs (odds ratios, OR = 3.92 and 3.25, *p* = 1.39 × 10*^−^*^5^, and 1.96 × 10*^−^*^4^ across two dcOrg replicates; Appendix A). On the other hand, human transcripts that were down-regulated by CVB infection in sc-islets remained proportionally unchanged for down-regulated transcripts via IAV infection in dcOrgs versus HSV-1 infection in dcOrgs (OR = 1.17 and 1.15, *p* = 0.57, and 0.67). These results indicated that there were more up-regulated human transcripts that were in common between the two RNA viruses (CVB and IAV) than between the RNA virus, CVB, and the DNA virus, HSV-1.

Down-regulated transcripts due to CVB infection in sc-islets were enriched for down-regulated transcripts by HSV-1 infection in sc-islets versus HSV-1 infection in dcOrgs (OR = 2.51 and 2.47, *p* = 2.13 × 10*^−^*^11^, and 1.14 × 10*^−^*^10^ across two dcOrg replicates). On the other hand, there were modest enrichments of up-regulated transcripts due to CVB infection in sc-islets that were in common with HSV-1 infection in sc-islets, in comparison to HSV-1 infection in dcOrgs (OR = 1.91 and 1.58, *p* = 3.07 × 10*^−^*^4^, and 0.013 across two dcOrg replicates). These results indicated that more down-regulated transcripts were in common between viral infections in the same cell culture system (sc-islets), compared to CVB and HSV-1.

Up-regulated transcripts via HSV-1 infection in dcOrgs had more transcripts in common with HSV-1 infection in sc-islets than CVB infection in sc-islets (OR = 6.67 and 5.41, *p* < 2.2 × 10*^−^*^16^ for both dcOrg replicates). Down-regulated transcripts via HSV-1 infection in dcOrgs did not have significantly enriched transcripts that were in common with HSV-1 infection in sc-islets versus CVB infection in sc-islets (OR = 0.82 and 0.87, *p* = 0.046, and 0.19 for both dcOrg replicates). These results indicated that more up-regulated transcripts were in common between HSV-1 infection across both cell culture systems (islets and dcOrgs). As baseline comparisons, we did not observe significant differences in the proportions of up-regulated or down-regulated transcripts across both dcOrg replicates (OR = 0.83, *p* = 0.32 and OR = 0.99, *p* = 0.95, respectively).

By using the percentage of transcripts that showed opposite directions as other measures for identifying conditions that are specific to each cell culture system, we observed that CVB-infected sc-islets and HSV-1-infected dcOrgs had the highest percentages of transcripts with discordant fold changes (Figure 5). Collectively, these results showed that CVB-infected sc-islets may be a superior model for studying transcriptomic perturbations that are specific to the system and, thus, they may be more relevant for T1D-associated processes. On the other hand, HSV-1-infected dcOrgs may be great models for studying transcriptomic perturbations that are specific to the system, and they may be more relevant to AD-associated processes.

The figure shows the percentages of DEGs that were up-regulated, down-regulated or had fold change differences in opposite directions in sc-islet infected by CVB or HSV-1 or dcOrgs infected by IAV or HSV-1, both of which were normalized by the total number of DEGs. The systems with the highest discordance (dark blue bars represent opposite directions), indicated by the yellow arrows, were CVB-infected sc-islets and HSV-1-infected dcOrgs.

Several interesting results were identified from our analyses. We can compare HSV-1-infected sc-islets to HSV-1-infected dcOrgs or IAV-infected dcOrgs to study up-regulated human transcripts that are in common (Figure 5). These transcripts may be relevant to inflammation across multiple tissues and organs. Similarly, we can compare CVB-infected sc-islets to HSV-1-infected sc-islets to study down-regulated human transcripts that are common, and these transcripts may be specifically relevant to inflammation in sc-islets.

## 4. Discussion

Virus-induced inflammation models using human embryonic stem cells or iPSC-derived dcOrgs and sc-islets have shown to be valuable tools for studying molecular transcriptomic signatures and processes that were associated with AD and T1D. Although AD and T1D are distinct diseases, there are interesting molecular similarities between AD and T1D. For instance, it has been consistently reported that innate immune activation in AD, leading to neuronal loss, is a key feature found in post-mortem brains of AD patients [45]. Similarly, inflammation leading to pancreatic beta-cell loss associated with T1D has been reported [46,47]. Moreover, beta-amyloid (Aβ) accumulation associated with AD can occur in both the brain and pancreas [48].

We previously found that the HSV-1 infection of dcOrgs led to transcriptomic perturbations associated with AD and that ACV treatment was sufficient to rescue these transcriptomic perturbations associated with AD in HSV-1-infected dcOrgs [4]. In this study, we found that HSV-1 infection of sc-islets led to transcriptomic perturbations associated with T1D and other autoimmune diseases. ACV treatment was insufficient to rescue these transcriptomic perturbations associated with T1D in HSV-1-infected sc-islets.

CVB-infected sc-islets and HSV-1-infected dcOrgs had the highest percentages of transcripts with discordant fold changes, indicating that we may be able to study specific inflammation-induced transcriptomic processes that are unique to each system and that these processes may inform us about disease-specific biology associated with T1D or AD. At the same time, we can use different viruses, such as RNA viruses (CVB, IAV) or DNA viruses (HSV-1) across both cell culture systems, to perturb transcripts that may inform us about the underlying common biology of inflammation-induced changes across multiple tissues or diseases.

It is also interesting to note that enteroviruses such as CVB had been associated with islet inflammation that was implicated in T1D [49], whereas herpesviruses such as HSV-1 had been associated with AD [50,51]. Our results indicate that while the use of anti-herpetic drug ACV can reverse AD-associated transcript perturbations in dcOrgs and thus ACV might reduce neuroinflammation-associated AD clinical pathology; however, ACV might not reduce inflammation-associated T1D clinical pathology. On the other hand, CVB vaccination may be able to prevent T1D, as reported in prior literature [52].

There are several future directions for our work, including direct comparisons of different DNA or RNA viruses (CVB, IAV, human cytomegalovirus, Zika virus, Sendai virus, HIV-1, SARS-CoV-2) in both cell culture systems. In addition, we can expand our work to explore the use of cytokines or DNA/RNA mimetics in inducing autoimmune transcripts. In our current work, we differentiated dcOrgs and sc-islets from individual donors. In the future, we can explore these neuroinflammation-induced effects using organoids from multiple donors to identify common processes across multiple donors. Single-cell RNA sequencing studies can also provide further delineation of cell types in which interesting transcripts are up-regulated or down-regulated in each system to enable insights into the initiation of autoimmunity involved in disease pathogenesis and progression.

## Figures and Tables

**Figure 1 cells-13-01978-f001:**
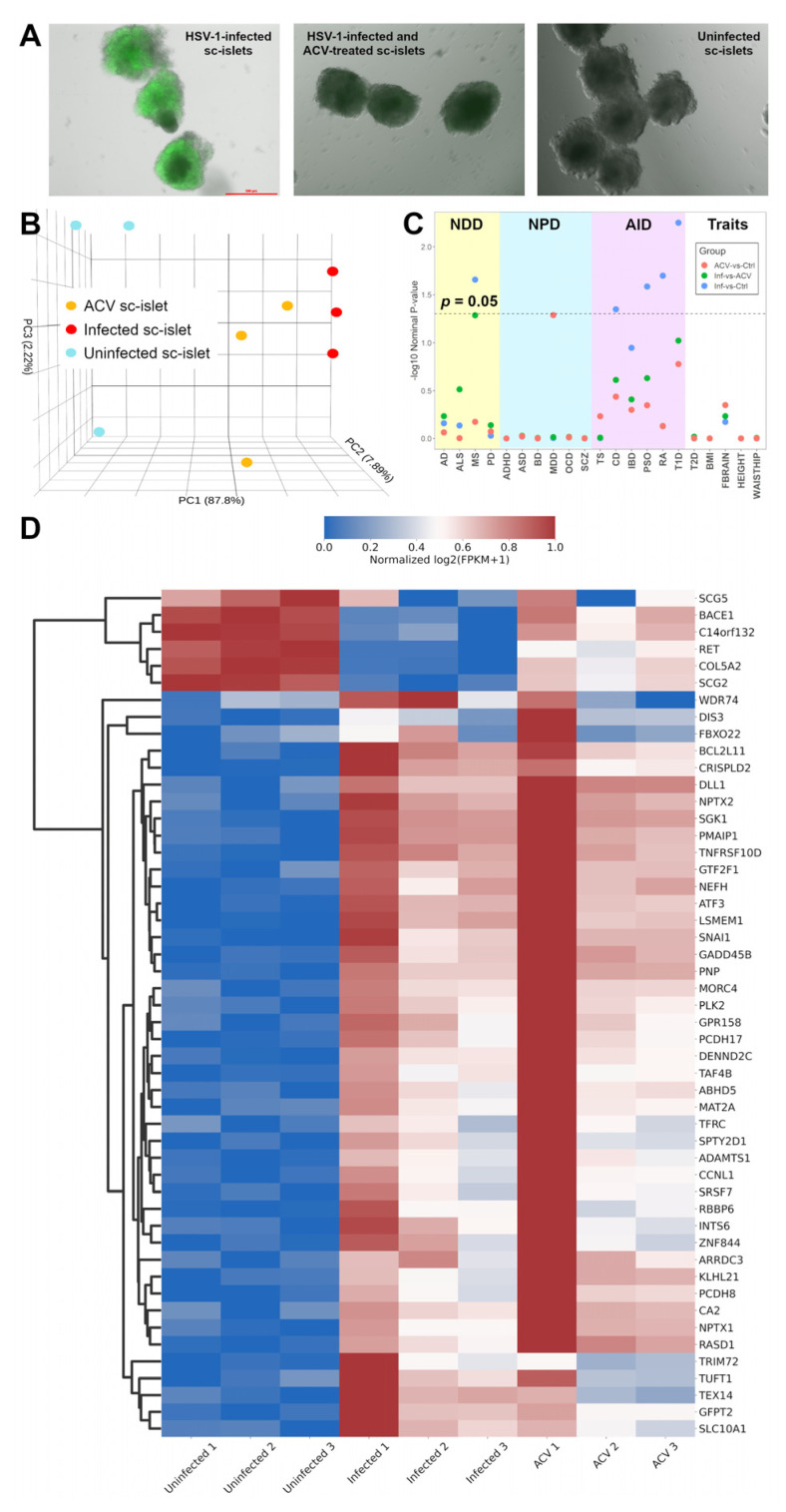
Analyses of RNA-seq data from sc-islets showed enrichment of autoimmune disease genes. (**A**) Images of unfixed HSV-1-infected sc-islets, HSV-1-infected and ACV-treated sc-islets, and uninfected sc-islets after 48 h. (**B**) PCA plot of the sc-islet RNA-seq data showing clustering of uninfected sc-islets (in blue), infected sc-islets (in red), and HSV-1-infected and ACV-treated sc-islets (in orange) along PC1 (*x*-axis). (**C**) Plot showing GSEA results from comparisons of the sc-islet RNA-seq datasets across 21 common diseases. The labels are “NDD” for neurodegenerative diseases, “NPD” for neuropsychiatric disorders, “AID” for autoimmune diseases, and “Traits” for traits and related diseases. The *y*-axis shows the -log10(nominal *p*-value from the GSEA result) and the *x*-axis shows each disease: AD for Alzheimer’s disease, ALS for amyotrophic lateral sclerosis, MS for multiple sclerosis, PD for Parkinson’s disease, ADHD for attention deficit hyperactivity disorder, ASD for autism spectrum disorder, BD for bipolar disorder, MDD for major depressive disorder, OCD for obsessive-compulsive disorder, SCZ for schizophrenia, TS for Tourette’s syndrome, CD for Crohn’s disease, IBD for inflammatory bowel disease, PSO for psoriasis, RA for rheumatoid arthritis, T1D for type 1 diabetes, T2D for type 2 diabetes, BMI for body mass index, FBRAIN for FMRI brain measurements, HEIGHT for height, WAISTHIP for waist-to-hip ratio. The comparison groups are: HSV-1-infected and ACV-treated versus uninfected sc-islets (ACV versus Ctrl, in red), HSV-1-infected versus HSV-1-infected and ACV-treated sc-islets (Inf versus ACV), and HSV-1-infected versus uninfected sc-islets (Inf versus Ctrl, in blue). (**D**) Heatmap showing the normalized log2 (FPKM+1) expression for the top 50 DEGs sorted by the adjusted *p*-values for uninfected sc-islets, infected sc-islets, and ACV-treated sc-islets. (**E**) Plot to show GSEA results from previously published RNA-seq datasets from HSV-1-infected 3D cerebral organoids. For the GSE145496 dataset, we used the 8dpi samples (mock versus HSV-1-infected) for the analyses.

**Figure 2 cells-13-01978-f002:**
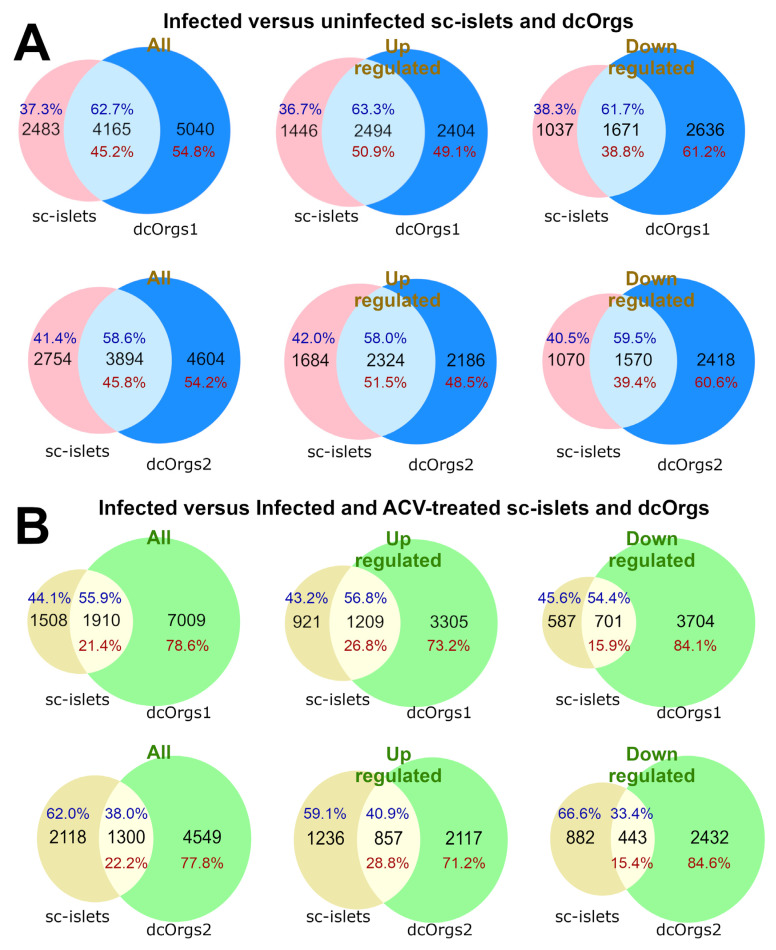
Comparisons of DEGs from HSV-1 infection in sc-islets versus two dcOrg replicates (dcOrgs1 and dcOrgs2). (**A**) Venn diagrams showing the numbers of HSV-1-infected versus uninfected DEGs that are common between sc-islets and dcOrgs1, as well as sc-islets and dcOrgs2. The percentages in blue text are calculated using the sc-islet DEGs, and the percentages in red text are calculated using the dcOrg DEGs. (**B**) Venn diagrams showing the numbers of HSV-1-infected versus HSV-1-infected and ACV-treated DEGs that are in common between sc-islets and dcOrgs1, as well as sc-islets and dcOrgs2. The percentages in blue text are calculated using the sc-islet DEGs, and the percentages in red text are calculated using the dcOrg DEGs.

**Figure 3 cells-13-01978-f003:**
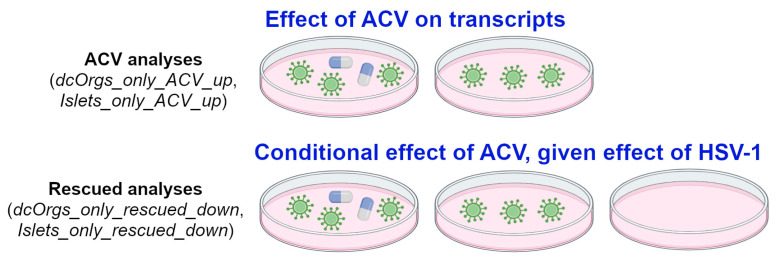
Schematic of our ACV and rescued analyses.

**Figure 4 cells-13-01978-f004:**
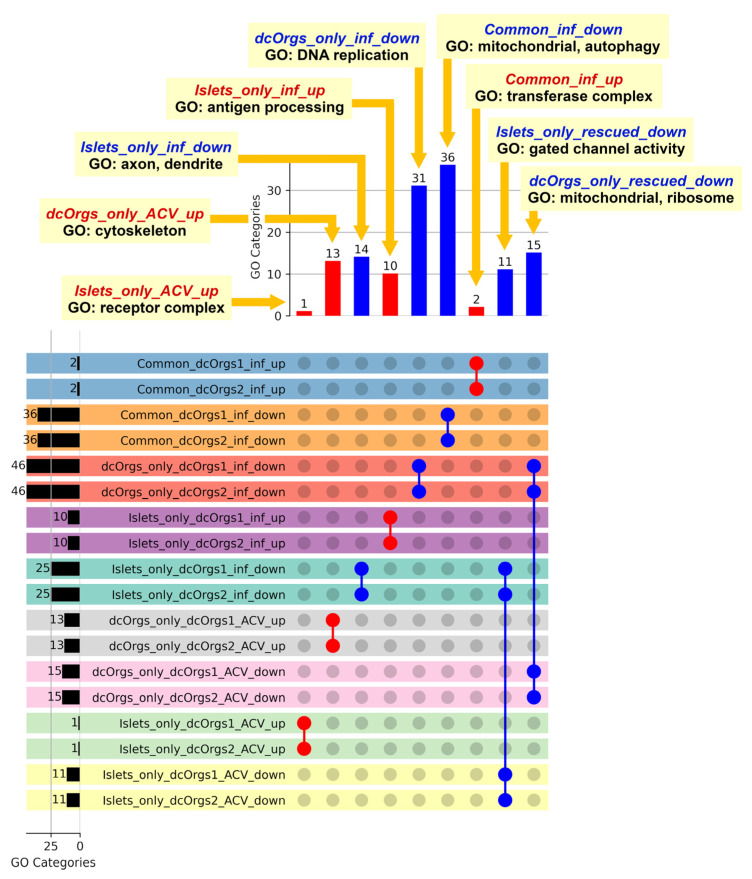
Shared GO categories across HSV-1-infected, or HSV-1-infected and ACV-treated sc-islets and dcOrgs. The bar graph on the left shows the total number of GO categories for each comparison. The bar graph at the top shows the numbers of shared GO categories across the DEG results, and the UpSet plot shows the DEG results that share these distinct GO categories. Shaded circles indicate the DEG result that shares the GO category; the red shaded circles highlight the up-regulated DEG results with the GO categories that were mentioned in the main text; the blue shaded circles highlight the down-regulated DEG results with the GO categories that were mentioned in the main text. The first four rows in the UpSet plot show the shared GO categories for the up-regulated genes in common due to HSV-1 infection in sc-islets and both sets of dcOrg replicates (Common_dcOrgs1_inf_up and Common_dcOrgs2_inf_up), down-regulated genes in common in HSV-1-infected sc-islets and both dcOrg replicates (Common_dcOrgs1_inf_down and Common_dcOrgs2_inf_down). The next eight rows in the UpSet plot show the shared GO categories for up-regulated or down-regulated genes exclusively in dcOrgs or exclusively in islets. The results for the HSV-1-infected and ACV-treated datasets are shown in the next 12 rows.

**Figure 5 cells-13-01978-f005:**
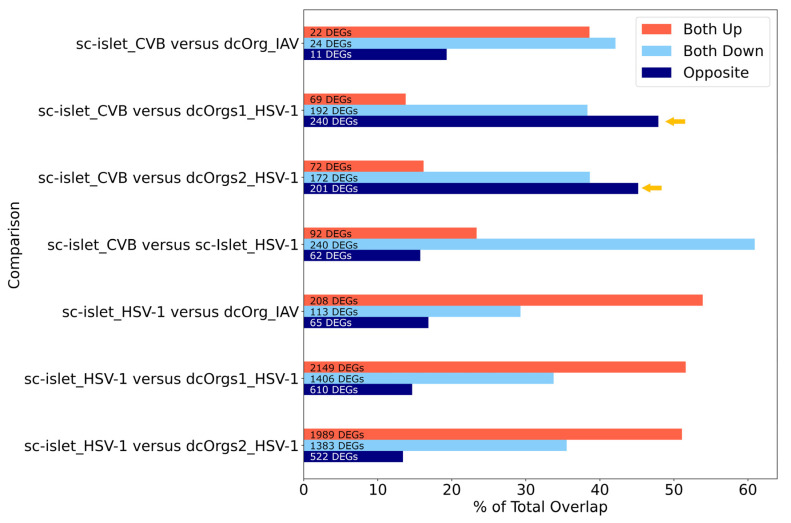
Comparisons of DEGs from CVB, IAV, and HSV-1 infection in sc-islets and dcOrgs revealed highest discordance between CVB-infected sc-islets and HSV-1-infected dcOrgs (highlighted using the yellow arrows).

## Data Availability

The RNA sequence data for the HSV-1-infected sc-islets have been deposited into GEO (accession GSE272361) and our scripts have been uploaded to GitHub (https://gitlab.com/elimlab/islets_dcOrgs accessed on 20 November 2024).

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
