# Peer review of "Herpes Simplex Virus 1 Infection of Human Brain Organoids and Pancreatic Stem Cell-Islets Drives Organoid-Specific Transcripts Associated with Alzheimer’s Disease and Autoimmune Diseases"

_cells, 2024, doi:10.3390/cells13231978_

Round 1

Reviewer 1 Report (Previous Reviewer 3)

Comments and Suggestions for Authors

The manuscript by Jonathan Sundstrom et al submitted to me for review
has been thoroughly revised. In my opinion,
it qualifies for publication in its current form.

Author Response

Thank you!

Reviewer 2 Report (New Reviewer)

Comments and Suggestions for Authors

In the manuscript submitted to me for review entitled "Herpes simplex virus 1 infection of human brain organoids and pancreatic stem cell-islets drives transcripts associated with Alzheimer’s disease and autoimmune diseases the authors Jonathan Sundstrom, Emma Vanderleeden, Nathaniel J Barton, Sambra D Redick, Pepper Dawes, Liam F Murray, Meagan N Olson, Khanh Tran, Samantha M Chigas, Adrian R Orszulak, George M Church, Benjamin Readhead, Hyung Suk Oh, David M Harlan, David M Knipe, Jennifer P Wang, Yingleong Chan and Elaine T Lim present a study identifying transcriptomic changes due to HSV-1 infection and acyclovir treatment upon HSV-1 infection in dissociated cells from human cerebral organoids (dcOrgs) versus stem cell-derived pancreatic islets (sc-islets). The results presented in the study could provide potential biological insight into the importance of HSV-1-induced inflammation in Alzheimer's disease (AD) and type 1 diabetes (T1D).

The methods presented by the authors are well chosen and described and can be replicated in a subsequent study. The obtained results are presented in detail using 5 figures in the main text of the manuscript and 1 supplementary figure and 9 supplementary tables presented in the supplementary file. To support their research, the authors used 51 references, with more than 2/5 of the total references being from the last 5 years. I did not notice any redundant self-citations, all references used are appropriate and necessary for the preparation of the manuscript.

My remarks and recommendations to the authors are:

1. Why was MOI = 5 used in "HSV-1 viral infection and ACV treatment of sc-islets"? The actual infection is at an MOI < 1, why was such an MOI chosen to conduct the experiment?

2. In Figure 1C, the captions inside the figure are too small and of poor quality. Even at very high magnification they are almost unreadable. If possible, improve their quality.

3. In the Materials and Methods section, only "HSV-1 viral infection" is indicated. In the "Results" section there is a subsection 3. 9. "Comparisons of transcriptomic perturbations by RNA and DNA viruses in sc-islets and dcOrgs revealed similarities in gene expression". The methodology of this experiment is not described in the Materials and Methods section. Figure 5 presents results from experiments performed with CVB and IAV. They are not at all mentioned and described in Materials and Methods. Please include information on the CVB and IAV strains used.

4. It is not a mandatory part of the manuscript, but many results are presented and it would be good if a brief conclusion was formulated.

5. Almost all references in the "References" section are presented with only first author and et al. As required by MDPI "et al." is placed if there are more than 10 authors. Let the missing authors be added. See instructions for authors.

·       For documents co-authored by a large number of persons (more than 10 authors), you can cite the first ten authors, then add a semicolon and add 'et al.' at the end:

Author 1; Author 2; Author 3; Author 4; Author 5; Author 6; Author 7; Author 8; Author 9; Author 10; et al.

Author Response

Comments 1: Why was MOI = 5 used in "HSV-1 viral infection and ACV treatment of sc-islets"? The actual infection is at an MOI < 1, why was such an MOI chosen to conduct the experiment?

Response 1: Thank you for pointing this out. We elaborated on this in the Methods section as follow: “We chose to use a MOI of 5 in our sc-islet model system to ensure that >50% of the cells would be infected and that we would observe significant transcriptomic perturbations after 48 hours. We would like to note that viral inflammation-induced disease pathology in humans is likely to be driven by low dosage exposure of viruses over several years.”

Comments 2. In Figure 1C, the captions inside the figure are too small and of poor quality. Even at very high magnification they are almost unreadable. If possible, improve their quality.

Response 2: Thank you for your suggestion. We have used shortforms and enlarged the text in the figure. We stated in the figure legend: “The labels are “NDD” for neurodegenerative diseases, “NPD” for neuropsychiatric disorders, “AID” for autoimmune diseases and “Traits” for traits and related diseases.”

Comments 3. In the Materials and Methods section, only "HSV-1 viral infection" is indicated. In the "Results" section there is a subsection 3. 9. "Comparisons of transcriptomic perturbations by RNA and DNA viruses in sc-islets and dcOrgs revealed similarities in gene expression". The methodology of this experiment is not described in the Materials and Methods section. Figure 5 presents results from experiments performed with CVB and IAV. They are not at all mentioned and described in Materials and Methods. Please include information on the CVB and IAV strains used.

Response 3: Thank you for pointing this out. We have now included information on the strains used: “We had also used influenza A (IAV) strain A/Puerto Rico/8/1934(H1N1) that was purchased from Charles River Laboratories [23], with a final HA titer per 0.05mL of 131,072 and the EID50 titer per mL was 109.8. The CVB strain used in our study was strain JVB (ATCC Cat #VR-184) [8].”

Comments 4. It is not a mandatory part of the manuscript, but many results are presented and it would be good if a brief conclusion was formulated.

Response 4: Thank you for the comment. We have added brief conclusions to several results paragraphs to explain the critical findings from the results.

Comments 5. Almost all references in the "References" section are presented with only first author and „et al“. As required by MDPI "et al." is placed if there are more than 10 authors. Let the missing authors be added. See instructions for authors.

  • For documents co-authored by a large number of persons (more than 10 authors), you can cite the first ten authors, then add a semicolon and add 'et al.' at the end:

Author 1; Author 2; Author 3; Author 4; Author 5; Author 6; Author 7; Author 8; Author 9; Author 10; et al.

Response 5: Thank you for pointing this out. We have now updated the references.

Reviewer 3 Report (New Reviewer)

Comments and Suggestions for Authors

The work by Sundstrom et al. evaluates the role of HSV-1 infection in the induction of inflammation in AD and T1D in two in vitro models. The approach that they use is the global transcriptomic profile of HSV-1-induced inflammation. The article has several flaws that make it impossible to accept for publication in the present form.

- The abstract must be improved to communicate the relevance of the virus and the studied diseases before mentioning the experimental design and results.

-The introduction fails to establish the relevance of the study. It is always necessary to describe the viral agent and the mentioned diseases. These points explain why it is essential to investigate this field. Additionally, the authors must explain (not with references) the advantages of these in vitro systems.

- Lines 69-71 are disconnected from the other paragraphs.

- Lines 72-91 and 92-99 must be rewritten since it is difficult to follow.

- The rationale for "infecting" sc-islets with HSV-1. Pancreatitis induced by this viral infection is rare and occurs in immunocompromised individuals. This point is not well supported.

- Line 200: All results must be shown.

- Line 274: All autoimmune diseases must be described.

-In general, the quality of the writing is poor and very convoluted. There is no real support for the study, and the results are not presented in a way that makes sense.

l

Author Response

Comments 1: The abstract must be improved to communicate the relevance of the virus and the studied diseases before mentioning the experimental design and results.

Response 1: Thank you for raising this point. We have added the following text in the abstract: “In addition, we compared transcriptomic signatures from HSV-1-infected sc-islets with sc-islets that were infected with coxsackie B virus (CVB) that had been associated with T1D pathogenesis.”

Comments 2: The introduction fails to establish the relevance of the study. It is always necessary to describe the viral agent and the mentioned diseases. These points explain why it is essential to investigate this field. Additionally, the authors must explain (not with references) the advantages of these in vitro systems.

Response 2: HUES8-derived SC-islets can be reproducibly generated from a single genetic background (HUES8 is among the first lines approved by NIH in response to a 2009 Executive Order enabling more hESC lines to be added to the list of those approved for federally funded research). They are accessible to other investigators. The SC-islets are derived from endoderm and thus lack macrophages that could confound innate immune responses in primary human islets. Similarly, cerebral organoids can be reproducibly generated from a single genetic background and are comprised of major cell types found in the human brain. In particular, the cerebral organoids used in our study had been reported to harbor microglia (that differentiated from the mesoderm lineage) innately within these organoids, thus providing a good human in-vitro model for studying innate immune response in the context of other cell types found in the central nervous system. We have added this text to our Methods section.

Comments 3: Lines 69-71 are disconnected from the other paragraphs.

Response 3: Thank you for pointing this out. We have now moved the text into Section 3.4 when we compared 3D versus 2D brain organoids.

Comments 4: Lines 72-91 and 92-99 must be rewritten since it is difficult to follow.

Response 4: Thank you for pointing this out. We have now rewrtitten the text as follow:

“Previously, we found that HSV-1 infection in dcOrgs led to several molecular and cellular phenotypes associated with AD neuropathology, including an enrichment of differentially expressed genes (DEGs) in AD-associated genes [4]. On the other hand, viral infections in dcOrgs by using the single-stranded RNA virus influenza A (IAV) did not result in these AD-associated phenotypes. This suggest that AD pathology can be activated by herpesviruses such as HSV-1. Interestingly, we also observed that HSV-1-infected dcOrg samples that were treated with the anti-viral drug acyclovir (ACV) had differential expression in transcripts associated with autoimmune diseases such as T1D and rheumatoid arthritis (RA). This indicates that the presence of HSV-1 viral con-structs may be sufficient to elucidate host inflammatory programs that are shared amongst different autoimmune diseases.

As such, we sought to test the hypothesis if HSV-1-infection in sc-islets can similarly lead to an enrichment of DEGs in AD-associated genes. If the hypothesis is true, the results will indicate that shared host inflammatory programs due to HSV-1 infection are driving disease-associated mechanisms in AD, regardless of the in-vitro human system (sc-islets or dcOrgs). On the other hand, if the hypothesis is not true, the results will indicate that there might be cell type-specific, or organoid-specific, host inflammatory programs that are driving disease-associated mechanisms in AD and T1D.

We further explore the global transcriptomic profile of HSV-1-induced inflammation in sc-islets and compared RNA sequencing (RNA-seq) data from HSV-1 infection (with and without ACV treatment) of sc-islets and dcOrgs to understand the perturbed gene networks and pathways that are in common or specific to each stem cell-derived system. Through our cross-organoid and cross-disease study, we aim to understand the specificity, efficacy and potential off-target effects of acute inflammation and anti-viral treatment, which can affect multiple organs physiologically in humans.”

Comments 5: The rationale for "infecting" sc-islets with HSV-1. Pancreatitis induced by this viral infection is rare and occurs in immunocompromised individuals. This point is not well supported.

Response 5: We use HSV-1 in SC-islets to interrogate how human islets respond to a virus that induces innate immune pathways distinct from that by CVB4.

Comments 6: Line 200: All results must be shown.

Response 6: Thank you for pointing this out. It was a mistake on our part as we had decided to leave out the stained images while we work on optimizing the staining protocol. We have now removed the section in the text.

Comments 7: Line 274: All autoimmune diseases must be described.

Response 7: Thank you for the suggestion. We had decided not to describe each autoimmune disease in depth as this may distract the readers from our main research focus in this manuscript on Alzheimer’s disease (AD) and type 1 diabetes (T1D).

Comments 8: In general, the quality of the writing is poor and very convoluted. There is no real support for the study, and the results are not presented in a way that makes sense.

Response 8: We thank the reviewer for the specific comments and have rewritten the manuscript. We hope that this revised manuscript is now presenting the results in a clearer manner.

Round 2

Reviewer 3 Report (New Reviewer)

Comments and Suggestions for Authors

The authors made changes only in what was written. However, the experimental design is questionable, and there were no changes in this regard or appropriate foundations to justify it.

This manuscript is a resubmission of an earlier submission. The following is a list of the peer review reports and author responses from that submission.

Round 1

Reviewer 1 Report

Comments and Suggestions for Authors
  • A summary

This study found that HSV-1-infected SC islets were enriched for autoimmune disease-associated genes, and ACV treatment showed little rescue effect. The study's outcomes provide new insights into the joint and unique transcriptomic response to HSV-1 virus by two cell models.

  • General concept comments

1.     What are the cut-offs used in this study to define a DEG? In Table S2, around 16K genes were detected, and the numbers for DEGs range from 3000-9000. Any comments on the DEGs numbers? Is it reasonable to have 50% of the whole transcriptome as DEGs? Such DEG analysis assumes that the RNA level should be the same between your samples. Usually, 1-5% of all genes detected are DEGs. If 50% of the genes are DEGs, the fundamental assumption is wrong, and a different method should be used to normalize the data before DEG analysis.

2.     Even ACV treatment showed little rescue effect on autoimmune disease-associated genes, but still more than 3000 DEGs from the Islet_Inf-vs-ACV comparison. What are the potential functions of those DEGs?  

  • Specific comments 

3.     Panel A of Figure 1 lacks two scale bars. Panel C of Figure 1 has different labels, but the labels for each group should be uniform throughout the paper, as should the color coding.

4.     The authors claimed that 80% of sc-islets were infected after 48 hours; however, in Figure 1B, the green-positive cell seems less than 80%. If the 80% number comes from quantifying different sc-islets, add the data to supplemental data.

5.     Adding a heatmap for top DEGs combining all comparisons to Figure 1 will catch the reader’s eye.

6.     Table S2: there are four sheets for different comparisons; what’s the meaning of HSV-1_Inf-vs-ACV? Theoretically, we expect three comparisons from three groups.

7.     In Figure 3, the upset plot has too many columns to focus on the marked columns. It’s better to show only the marked columns and put the vast figure 3 in supplemental materials.

8.     Change the order of Figures 3 and 4 if Figure 3 shows the terms from Figure 4.

9.     Figure 5 shows the percentage of the overlapped regulated DEGs; please add the total overlapped number. How about the similarity between dcOrgs1 and dcOrgs2? Can this analysis be shown in Figure 5 as a baseline for the similarity?

10.   Line 508: If no iPSCs were used, please remove the PGP1 iPSCs from the acknowledgments.

Reviewer 2 Report

Comments and Suggestions for Authors

Herpes simplex virus 1 infection of human brain organoids and 2

pancreatic stem cell-islets drives transcripts associated with 3

Alzheimer’s disease and autoimmune diseases

CELL

Thank you for inviting me to review the above-titled manuscript. The issue of Alzheimer's disease is essential. However, the paper is not based on solid scientific methodology or statistical analysis.

Title: Too broad. May be suitable for newspapers and lacked focus. The author talk about 5 different issues. What is the focus of the research?

Abstract: 1) What is the research question? 2) What is the focus of the study? 3) What are you trying to answer or prove? 4) What type of research methods did you use? What are the tools? 5) The authors jump from one issue (a field) to another (field). 6) What statistical tests did you use, what were your measures - mean±SD, median, IQR, CI, p-values. 7) Conclusions not based on solid evidence.

Introduction- 1) What is the problem? What triggered the study? 2) What is the research question? What is the focus of the study? 3) what was the specific gap in the literature that you are looking at? What other researchers found? 

Methods- 1) You need a study design section to explain how you will answer your research question. 2) Some statements need references/citations. 3) How were the data processed? What were your statistical tests? Why? 

Results- I cannot see an analysis of data. 2) Figure 5- why the % were compared, and not mean±SD, and p-values shown. 3)( Figure 4- Does this figure show results? Are these results? 4) Figure 3 - were is the analysis? 

Discussion and conclusionThe argument of the authors is hypothetical and not based on solid evidence. The authors should have a focus for their research. 

References: Again, the focus is lacking, and this needs to be improved. 

Comments on the Quality of English Language

Herpes simplex virus 1 infection of human brain organoids and 2

pancreatic stem cell-islets drives transcripts associated with 3

Alzheimer’s disease and autoimmune diseases

CELL

Thank you for inviting me to review the above-titled manuscript. The issue of Alzheimer's disease is essential. However, the paper is not based on solid scientific methodology or statistical analysis.

Title: Too broad. May be suitable for newspapers and lacked focus. The author talk about 5 different issues. What is the focus of the research?

Abstract: 1) What is the research question? 2) What is the focus of the study? 3) What are you trying to answer or prove? 4) What type of research methods did you use? What are the tools? 5) The authors jump from one issue (a field) to another (field). 6) What statistical tests did you use, what were your measures - mean±SD, median, IQR, CI, p-values. 7) Conclusions not based on solid evidence.

Introduction- 1) What is the problem? What triggered the study? 2) What is the research question? What is the focus of the study? 3) what was the specific gap in the literature that you are looking at? What other researchers found? 

Methods- 1) You need a study design section to explain how you will answer your research question. 2) Some statements need references/citations. 3) How were the data processed? What were your statistical tests? Why? 

Results- I cannot see an analysis of data. 2) Figure 5- why the % were compared, and not mean±SD, and p-values shown. 3)( Figure 4- Does this figure show results? Are these results? 4) Figure 3 - were is the analysis? 

Discussion and conclusionThe argument of the authors is hypothetical and not based on solid evidence. The authors should have a focus for their research. 

References: Again, the focus is lacking, and this needs to be improved. 

Reviewer 3 Report

Comments and Suggestions for Authors “Herpes simplex virus 1 infection of human brain organoids and pancreatic stem cell-islets drives transcripts associated with  Alzheimer’s disease and autoimmune diseases” by Jonathan Sundstrom et al. is very interesting research project.

Research by various authors indicates that human embryonic stem cells or iPSCs dcOrgs derivatives and sc islets may be useful for investigating molecular transcriptomic signatures and processes associated with AD and T1D. Although AD and T1D are different diseases, there are significant molecular similarities between them. 

The subject of the study was the identification of transcriptomic changes due to HSV-1 infection and antiviral drug treatment (acyclovir, ACV) HSV-1 infection in separate human brain organoid cells (dcOrgs) compared with stem cell-derived pancreatic islets (sc-islets).  In the presented study, the authors observed that HSV-1 infection of sc islets led to transcriptomic disturbances. 

In the future, the Authors plan to compare different DNA or RNA viruses (CVB, IAV, human cytomegalovirus, Zika virus, Sendai virus, HIV-1, SARS-CoV-2) in both cell culture systems. Additionally, they plan explore the use of cytokines or DNA/RNA mimetics in inducing autoimmune transcripts. 

         In my opinion, the experimental part was carried out precisely using various methods. The results were presented very precisely. The value of the manuscript is increased by bar graph and figures. The authors believe that this model can be used to understand the mechanism of many viruses’ infection, including both pathogenesis and progression.

Comment: I think it would be good to explain whether the results obtained can have any application in clinical practice.

Reviewer 4 Report

Comments and Suggestions for Authors

Peer review report on ‘Herpes simplex virus 1 infection of human brain organoids and pancreatic stem cell-islets drives transcripts associated with Alzheimer’s disease and autoimmune diseases.’

Manuscript ID: Cells-3128959

This paper investigates the connection between HSV-1 infection and the development of Alzheimer’s disease (AD) and autoimmune diseases. The authors interrogated the result of infection in both brain organoids and pancreatic stem cell-islets (sc-islets) and illustrated convincingly a link between the two. Interestingly, HSV-1 infection in the brain organoids was associated exclusively with the development of AD whereas sc-islets demonstrated enrichment for genes associated with autoimmune diseases  like type 1 diabetes. Moreover, the authors conclude that AD pathology is not activated by all viruses but shows specificity. In contrast to previous work which indicated that Acyclovir (ACV) treatment showed potential to rescue cells after infection, disappointingly, this study found that ACV had no effect.

This study is very topical and with a burgeoning worldwide epidemic of AD and diseases like type 1 diabetes emerging, with an abundance of HSV-1 thrown into the mix it reveals serious implications for our health if we can’t control HSV-1, but equally brings some hope that these diseases may be managed. The paper is complex but well written and well described. The methods and experimental procedures are comprehensive, and the figures are clear.

Some comments:

It would be easier to read if the figures were moved so that they are interspersed with the appropriate text instead of being all grouped together. With the complexity of the subject the reader by necessity is required to move focus frequently from text to illustration. This becomes exceptionally tedious when the figures follow in a separate section.

In the materials and methods section, it is written that ACV 200 µM was added to various samples. How much was added?

You state that ACV demonstrated no ability to rescue those cells infected with HSV-1 in contrast to your previous research. However, even though as you say, the dosage of ACV may have been inadequate, if we disregard the tyranny of the P value for a moment, Figure 1D shows a distinct differential in the pattern of distribution in the active samples between ACV-vs-Ctrl, Inf-vs-ACV, and Inf-vs-Ctrl. This differential would indeed be consistent with your previous report. Could you comment on this.

Figures 1C and 1D do not have captions.

Please format the section numbering and headings consistently. Section 2 is all bold and section 3 is all italics.
